# Land–Lake Linkage and Remote Sensing Application in Water Quality Monitoring in Lake Okeechobee, Florida, USA

**Mohammad Hajigholizadeh [1]** , **Angelica Moncada [2]** , **Samuel Kent [2]** and **Assefa M. Melesse [2,*]**

1 Department of Civil & Environmental Engineering, Florida International University, 10555 W Flagler Street, EC3781, Miami, FL 33174, USA; mhaji002@fiu.edu
2 Department of Earth and Environment, Institute of Environment, Florida International University, AHC-5-390, 11200 SW 8th Street, Miami, FL 33199, USA; amonc002@fiu.edu (A.M.); skent@fiu.edu (S.K.)
* Correspondence: melessea@fiu.edu

**Abstract:** The state of water quality of lakes is highly related to watershed processes which will be responsible for the delivery of sediment, nutrients, and other pollutants to receiving water bodies. The spatiotemporal variability of water quality parameters along with the seasonal changes were studied for Lake Okeechobee, South Florida. The dynamics of selected four water quality parameters: total phosphate (TP), total Kjeldahl nitrogen (TKN), total suspended solid (TSS), and chlorophyll-a (chl-a) were analyzed using data from satellites and water quality monitoring stations. Statistical approaches were used to establish correlation between reflectance and observed water quality records. Landsat Thematic Mapper (TM) data (2000 and 2007) and Landsat Operational Land Imager (OLI) in 2015 in dry and wet seasons were used in the analysis of water quality variability in Lake Okeechobee. Water quality parameters were collected from twenty-six (26) monitoring stations for model development and validation. In the regression model developed, individual bands, band ratios and various combination of bands were used to establish correlation, and hence generate the models. A stepwise multiple linear regression (MLR) approach was employed and the results showed that for the dry season, higher coefficient of determination ($R^2$) were found ($R^2 = 0.84$ for chl-a and $R^2 = 0.67$ for TSS) between observed water quality data and the reflectance data from the remotely-sensed data. For the wet season, the $R^2$ values were moderate ($R^2 = 0.48$ for chl-a and $R^2 = 0.60$ for TSS). It was also found that strong correlation was found for TP and TKN with chl-a, TSS, and selected band ratios. Total phosphate and TKN were estimated using best-fit multiple linear regression models as a function of reflectance data from Landsat TM and OLI, and ground data. This analysis showed a high coefficient of determination in dry season ($R^2 = 0.92$ for TP and $R^2 = 0.94$ for TKN) and in wet season ($R^2 = 0.89$ for TP and $R^2 = 0.93$ for TKN). Based on the findings, the Multiple linear regression (MLR) model can be a useful tool for monitoring large lakes like Lake Okeechobee and also predict the spatiotemporal variability of both optically active (Chl-a and TSS) and inactive water (nutrients) quality parameters

**Keywords:** Lake Okeechobee; Landsat; multiple linear regression; chlorophyll-a; total suspended solids; nutrients





## 1. Introduction

Water quality indicators including physical, chemical, and biological properties are traditionally determined by analyzing water samples from the field. Although this method offers accurate results, it has proven to be labor intensive, time-consuming, and expensive. Analyses are limited to a single point in space and time, which is challenging when these values are vital for watershed assessments and management practices. To bridge the gap between accuracy and large-scale analyses, remote sensing technology has demonstrated to be a great tool. Using remote sensing techniques outlying regions and waterbodies can be monitored more effectively. Remotely sensed data in digital form is easily acquired and processed and it has been used since the 1970s for water quality assessments in the developed world [1–16].

Advanced geostatistical techniques are gaining support among researchers for spatiotemporal evaluation of surface water quality when space/time monitoring data are sparse. However, data collection is an expensive and time-consuming process. Spatial and temporal evaluation of surface water quality parameters (WQPs) and quantitative study of changes can also be performed in a more effective and efficient manner using remote sensing. As most studies focus on improving quantification methods for optically active variables such as chlorophyll-a (chl-a), total suspended solids (TSS), and turbidity, there is a growing need for quantifying variables with weak optical characteristics and low signal noise ratio such as pH, total nitrogen (TN), ammonia ($NH_3$-N), nitrate ($NO_3$-N), and dissolved phosphorus (DP). Although difficult to quantify, these parameters are important for water quality assessments; therefore, any significant finding in this field will increase the applicability of remote sensing technology. This study aims to quantify these variables by establishing significant correlations between satellite data and in-situ measurements of both optically strong and weak WQPs in a freshwater lake.

It is important to analyze both optically strong and weak parameters in the same study in order to have a good reference from established methodology while also looking at new possible correlations. The WQPs that are studied using remote sensing techniques are well established in the lake ecology literature for water quality assessment. These parameters include Chlorophyll-a (chl-a), total suspended solids, phosphorus, and nitrogen. The first two are optically strong and the latter two are not well studied in the remote sensing literature because they are optically weak, so they are not readily detected with current satellite technology. Nevertheless, all of the parameters are needed for well-rounded water quality assessments, so establishing correlations between them is important.

Chl-a is used as an indicator of a waterbody's trophic state and is used as a proxy for net primary production and algal blooms on surface water which are enhanced with eutrophication. To quantify chl-a from satellite data, different spectral bands and various combination and ratios are considered and tested. Spectral band ratios can reduce irradiance and atmospheric and air-water surface influences in the signal [17,18]. There is good evidence that Landsat visible bands are appropriate for detecting chlorophyll-a concentration in lake water. For example, Alparslan et al. [19] utilized all bands of Landsat-5 TM to estimate the concentration of chl-a and figured out there were significant correlation between different TM bands and chl-a concentrations. Landsat-8/OLI images were used by Lim and Choi [20] and found that a significant correlation between chl-a concentration and OLI bands and their ratios with obtained R values for bands 2, 3, 4 and band 5/band 3 equal to −0.66, −0.70, −0.64, and −0.64, respectively. OLI bands 1 through band 4 and their ratios were also found correlated with chl-a concentration by another study implemented by Zhang and Han [21]. Kim et al. [22] also used Band 2, 5, and a ratio of Band 2/Band 4 of Landsat-8/OLI to investigate the concentration of chl-a.

TSS is the measure of particles in the water column that carry many of the pollutants and sediment. By measuring TSS in a water body, managers can see how much of these particles it carries. As suspended particles increase in a waterbody, light would be more scattered and harder to travel through the water column, and therefore the water's turbidity index will be higher. TSS is responsible for most of the scattering and chl-a and colored dissolved particles mainly control the absorption of light in surface waterbodies [23]. In theory, the use of a single band should provide a robust and TSS-sensitive algorithm when an appropriate band is chosen [24]. Curran et al. [25] and Novo et al. [26] showed that single band algorithms may be adopted where TSS increases with increasing reflectance; however, the complex substances in water change the reflectance of the water body and therefore cause variation in colors, and thus, the use of different spectral bands can be more appropriate [24,27,28]. Ritchie et al. [29] backed up by in-situ studies show that the most useful range of spectrum for the determination of suspended particles in surface waters was between 700 and 800 nm. In the Near-IR and Mid-IR regions, water increasingly absorbs the light and makes it look darker, which varies based on water depth and wavelength.

Total phosphorus (TP), which is an optically weak parameter to be monitored by the application of remote sensing, is defined as the measurement of all forms of organic, inorganic, and dissolved phosphorus and is typically found as Phosphates ($PO_4$) in nature. Total phosphate is related to Secchi disk transparency (SDT) with an exponential equation which is consistent with Carlson's finding [1,30–32]. They are considered as one of the main nutrients required for plant growth, and their increase in waterbodies leads to increase in the rapid growth of algae. Although total phosphorus is very unlikely to be measured directly using remote sensing techniques. However, due to having a general correlation with other water quality parameters, total phosphorus can be studied indirectly by finding its relationship with other parameters. Landsat TM data have been widely used to evaluate the spatial and temporal pattern of TP [30–32] and empirical estimations and regression models were used to find the best correlations between TP concentration and other water quality parameters, such as chl-a concentration and Secchi Disk Depth (SDD) and Song et al. [33] found a good correlation of 0.62, 0.59, 0.55, and 0.51 between TP and TM1 (TM Band 1), TM2, TM3, and TM4, respectively. A combination of TM1, TM3/TM2, and TM1/TM3 data was used by Wu et al. [32] to correlate TP concentration with chl-a and Secchi Disk. Further, in another study on Ömerli Dam, Turkey, the first four bands of Landsat 7-ETM were used to estimate the concentration of TP by Alparslan et al. [1]. Alparslan et al. [19] later used Band1 to Band5, and Band7 of Landsat-5 TM to measure the concentration of total phosphorus. Landsat-8/OLI bands 2 to Band 5 were used by Lim and Choi [20] to build three different regression models testing single bands and their ratios and significant coefficient of determination values were found and reported.

Another weakly optical detectable water quality indicator is Total Kjeldahl Nitrogen (TKN) [34]. TKN is a measure of the amount of organic nitrogen and ammonia-N that due to adding nitrogen to water can lead to increase in the levels of chl-a and algae. Regardless, anthropogenic activities have greatly increased the availability of this nutrient in freshwater, so it is important to monitor as well. The organic matter in agricultural fertilizer released into waterbodies and its decomposition is reported to be the main source of TKN in most studied watersheds [35]. Different types of agricultural areas in South Florida, for example, generate non-point sources of pollution that have a high percentage of ammonia from pesticides and fertilizers [36]. Most suspended solids are made up of inorganic materials, though bacteria and algae can also contribute to the total solids' concentration. Organic particles from decomposing materials can also contribute to the TSS concentration.

There are not many studies that can clearly prove the application of remote sensing in measuring the concentration of nitrogen in waterbodies. However, Hood et al. [37] found that the nitrogen concentration is highly correlated to absorption and fluorescence characteristics of chlorophyll-a (chl-a). Another study by Hanson et al. [38] reported that chl-a fluorescence is influenced by TN and TP. Edwards et al. [39] showed that colored dissolved organic matters (CDOM), chl-a, and suspended sediment concentration (SSC) are strongly correlated with the level of nutrients in waterbodies. Hyperspectral remote sensing technique was applied by Gong et al. [40] to measure different concentrations of TN and TP reflectance spectra in the laboratory and under pure water condition to discover their special features. Results showed reflectance peaks at 404 and 477 nm for nitrogen, and reflectance peaks at 350 nm for phosphorus, and derived retrieval models for the measurement of nitrogen and phosphorus concentrations. Landsat 7/ETM+ data was used to study the concentration of nitrite nitrogen ($NO_2$-N) and nitrate nitrogen ($NO_3$-N) in Mersin Bay, Turkey [41], and MLR models were used to select the best bands to measure these water quality parameters.

Monitoring these contaminants in a lake environment with satellite technology is useful and important for the management of freshwater systems. Lake Okeechobee, in the Florida peninsula, has become a eutrophic system because of high nutrient loads from numerous sources including various types of agriculture, dairy and cattle pastures, sugarcane plantations, orange groves, and other plantations. The quality of water that flows into the lake will eventually make its way into the freshwater Everglades or its coastal

estuaries; thus, control of nutrient loading to Lake Okeechobee is a pivotal issue. In this study, the spatiotemporal changes of the four previously mentioned WQPs, including chl-a, total phosphate (TP), total suspended solids (TSS), and TKN, were studied using remote sensing, water quality monitoring stations' data, and statistical models. 26 monitoring stations' data were used to develop correlation analysis between optical bands from blue to near infrared and different band ratios, and also for the validation of the resulted regression models.

## 2. Materials and Methods

### 2.1. Study Area

South Florida is the only part of the United States with a subtropical climate, and Lake Okeechobee is the largest freshwater lake in Florida, United States. Figure 1 shows the location of the study area, Lake Okeechobee, and the selected water quality monitoring sites. It is a dynamic and productive system with various water inputs including the Kissimmee River in the north, a system of pumping stations on its south shore, and seasonal rains. During the wet periods various pumps divert untreated, surplus water from the Everglades agricultural area from the south into the lake. The Lake's watershed's main land uses are agriculture and urban which have high nutrient loading potential. The average annual temperature ranges from 19.2 °C to 28.7 °C and the annual rainfall in the entire area of South Florida is generally about 55 inches (1400 mm). Considering the subtropical climate of South Florida, the average rainfall is still considerable in the dry season. In addition, during El Niño phenomenon, Florida sees greater rainfall between November and March (dry season).

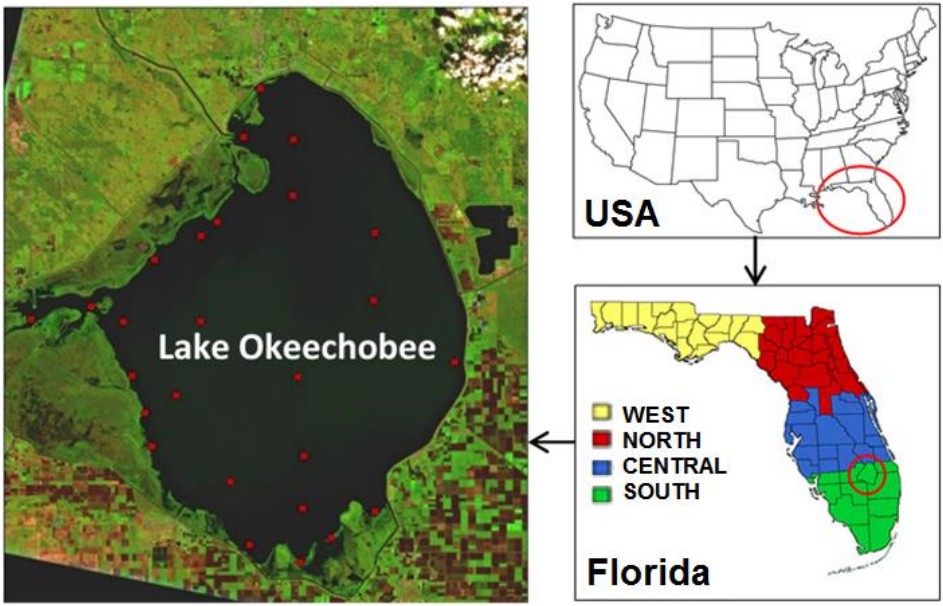

**Figure 1.** Study area and the water quality monitoring sites.

### 2.2. Datasets

#### 2.2.1. Limnological Data

Twenty-six (26) water quality monitoring stations located on Lake Okeechobee were downloaded from GIS data catalog of the South Florida Water Management District (SFWMD: https://data-swfwmd.opendata.arcgis.com/). These stations were selected on the basis of data continuity, distribution and availability of the selected parameters for the period of the study. Concentrations of chl-a, total phosphate, total suspended solids, and TKN were obtained from these stations for dates matching the imagery acquisition. Since TSS, SDT, and TP has been reported to be correlated [42–45], this study will explore to use a similar approach in developing a relationship between TP and SDT and TSS.

Availability of sampling data was considered in choosing satellite images, which also were chosen based on the two wet and dry seasons of South Florida. In this region, 15 May to 15 October is considered as wet season and from 16 October to 14 May is known as dry season. Using a random split, 75% of the in-situ data were used for the development of statistical model and their correlation with optical bands of imagery and 25% of data was used for the validation of the models. Data analysis was performed using SPSS 16.0 software package. The descriptive statistics of the four considered limnological parameters in this study are summarized in Table 1.

**Table 1.** Descriptive statistics of observed water quality parameters from the 26 stations of Lake Okeechobee. The dates correspond to the satellite imagery acquisition.

| | | | Chl-a (mg/m$^3$) | TSS (mg/L) | TP (mg/L) | TKN (mg/L) |
|---|---|---|---|---|---|---|
| Dry Season | 29/02/2000 | Min | 3.30 | 8.0 | 0.107 | 1.03 |
| | | Max | 67.7 | 53.0 | 0.192 | 1.96 |
| | | Mean | 17.8 | 29.5 | 0.156 | 1.31 |
| | | STD | 14.8 | 13.0 | 0.021 | 0.21 |
| | 31/01/2007 | Min | 1.0 | 8.0 | 0.085 | 0.88 |
| | | Max | 27.5 | 162.0 | 0.299 | 2.04 |
| | | Mean | 12.9 | 41.2 | 0.158 | 1.22 |
| | | STD | 7.1 | 36.0 | 0.050 | 0.30 |
| | 06/02/2015 | Min | 1.86 | 5.0 | 0.0821 | 0.821 |
| | | Max | 43.2 | 92.0 | 0.203 | 1.85 |
| | | Mean | 14.08 | 31.04 | 0.143 | 1.09 |
| | | STD | 8.38 | 21.54 | 0.033 | 0.22 |
| Wet Season | 06/07/2000 | Min | 6.20 | 3.0 | 0.036 | 0.95 |
| | | Max | 106.5 | 70.0 | 0.238 | 2.89 |
| | | Mean | 27.7 | 20.6 | 0.101 | 1.38 |
| | | STD | 23.4 | 15.3 | 0.044 | 0.48 |
| | 22/08/2007 | Min | 3.50 | 4.0 | 0.016 | 0.87 |
| | | Max | 76.5 | 22.0 | 0.280 | 2.61 |
| | | Mean | 16.1 | 11.3 | 0.092 | 1.35 |
| | | STD | 16.0 | 7.0 | 0.064 | 0.37 |
| | 15/09/2015 | Min | 4.80 | 3.0 | 0.023 | 0.77 |
| | | Max | 76.0 | 39.0 | 0.295 | 2.93 |
| | | Mean | 18.4 | 13.7 | 0.084 | 1.57 |
| | | STD | 16.8 | 9.3 | 0.047 | 0.34 |

Prior to the statistical analyses, different required data pre-treatment methods including missing data treatment, non-informative variables elimination, and outliers detection and required adjustment were carried out. Boxplot were built to explore the outliers. The values above or below the upper or lower fences that represent values more and less than 75th and 25th percentiles (3rd and 1st quartiles), respectively, were detected and removed.

### 2.2.2. Satellite Data

Landsat 5 Thematic Mapper (TM) and Landsat 8 Operational Land Imager OLI data in three dates of year 2000, 2007, and 2015 downloaded from United States Geological Survey (USGS) website. The Landsat data were geometrically corrected raw digital number with Level 1 correction. Thematic Mapper (TM) data from two dates (29 February 2000 and 31 January 2007), and Landsat Operational Land Imager (OLI) data in one date (6 February 2015) were obtained for dry season. TM data from two dates of (6 July 2000 and 11 August 2007), and OLI data from one date (15 September 2015) were obtained for wet season of South Florida to evaluate the temporal and spatial patterns of considered water quality parameters in Lake Okeechobee. Pre-processing of the Landsat data consisted of radiometric calibration and atmospheric correction which is necessary for quantitative

studies [46,47]. Pre-processing is especially important in the case of surfaces such as lake waters where the reflected light is small [48] resulting in a reflectance of less than 10% on average, and often lower than 1%. Digital number values of images were converted to unitless planetary reflectance to eliminate the effects of high local variability on remote sensing observation values, and separate methodologies were utilized for the atmospheric correction and radiometric calibration of TM and OLI data, which are described in detail in the methodology section. Image processing was performed using ERDAS IMAGINE version 2014 and ESRI ArcGIS version 10.0.

*2.3. Methodology*

2.3.1. Data Preprocessing: Landsat-5/TM

To remove the voltage bias and gains from the satellite sensor, the digital number (DN) values converted to radiance values for each band as follows:

$$L\lambda = \frac{(L_{max}\lambda - L_{min}\lambda)}{(QCAL_{max} - QCAL_{min})} \times (QCAL - QCAL_{min}) + L_{min}\lambda \tag{1}$$

where $L\lambda$ = spectral radiance at the sensor's aperture in watts/(meter squared $\times$ ster $\times$ μm), $L_{min}\lambda$ = the spectral radiance that is scaled to $QCAL_{min}$ in watts/(meter squared $\times$ ster $\times$ μm), $L_{max}\lambda$ = the spectral radiance that is scaled to $QCAL_{max}$ in watts/(meter squared $\times$ ster $\times$ μm), $QCAL_{min}$ = the minimum quantized calibrated pixel value (corresponding to $L_{min}\lambda$) in DN, and $QCAL_{max}$ = and the maximum quantized calibrated pixel value (corresponding to $L_{max}\lambda$) in DN.

As the sun angle varies in different latitude, time of day, season, and also due to different distances between the earth and sun, the radiance values should be converted to at-satellite reflectance values. Equation (2) shows the final reflectance determined using a simplified model of atmosphere effects [49]:

$$\rho p = \frac{\pi \times L_\lambda \times d^2}{ESUN_\lambda \times \cos\theta_s} \tag{2}$$

where $\rho p$ = unitless planetary reflectance, $L_\lambda$ = spectral radiance at the sensor's aperture, d = the Earth − Sun distance in astronomical units, $ESUN_\lambda$ = the mean solar exo-atmospheric irradiance, and $\theta_s$ = the solar zenith angle in degrees.

2.3.2. Data Preprocessing: Landsat-8/OLI

For each band of Landsat-8/OLI data, the digital number (DN) values were also converted to reflectance values using the improved cosine of the solar zenith angle (COST) method proposed by Moran et al. [50] as follows:

$$P\lambda = \frac{\pi (L_{\lambda sensor} - L_{\lambda haze})d^2}{ESUN_\lambda \cos\left(\frac{\pi}{180\theta_s}\right)} \tag{3}$$

where $P\lambda$ = the dimensionless spectral reflectance value of surface water, $\pi$ = a constant (3.14159265), $L_{\lambda sensor}$ = the spectral radiance value, $L_{\lambda haze}$ = the path radiance or upwelling atmospheric spectral radiance, d = the distance between the earth and the sun in astronomical units, $\theta_s$ = the solar zenith angle (°), and $ESUN_\lambda$ = the solar spectral irradiance at to the top of atmosphere (TOA).

The spectral radiance value ($L\lambda_{sensor}$) at the satellite sensor's aperture (Wm − 2sr − 1 μm − 1) is calculated as follows:

$$L\lambda_{sensor} = (M\lambda \times Q_{cal}) + A\lambda \tag{4}$$

where $M\lambda$ = the band-specific multiplicative rescaling factor, $A\lambda$ = the band-specific additive rescaling factor, and $Q_{cal}$ = the minimum quantized and calibrated standard product pixel

value. Both Mλ and Aλ are provided in the Landsat 8 metadata file (MTL file). The path radiance (i.e., the upwelling atmospheric spectral radiance scattered in the direction of the sensor entrance pupil and within the sensor's field of view), $L\lambda_{haze}$, is calculated as follows:

$$L\lambda_{haze} = L\lambda_{min} - L\lambda, 1\% \tag{5}$$

where $L\lambda_{haze}$ = the path radiance or upwelling atmospheric spectral radiance scattered in the direction of the sensor entrance pupil and within the sensor's field of view, $L\lambda_{min}$ = the minimum spectral radiance, and $L\lambda, 1\%$ = the spectral radiance value of the darkest object on each band of the Landsat 8 and is given by

$$L\lambda, 1\% = \frac{0.01 \times ESUN_{\lambda} \times \cos(\theta)^2}{\pi \times d^2} \tag{6}$$

Assuming that dark objects have 1% or smaller reflectance, the theoretical radiance of these object was then computed, accordingly [33,50].

### 2.3.3. Statistical Analyses

Commonly used statistical techniques were employed to determine the relationship between electromagnetic energy and water quality parameters [1,20,51–55]. The chosen parameters for analysis were Chl-a, TSS, total phosphorus, and TKN.

Studies have shown that there is a significant correlation between the visible bands and water transparency and chl-a concentration [51–53,55–58] and also between the visible bands of Landsat and total suspended matters [57,59–61].

Thus, Pearson's correlation analysis was used in this study to find the existing linear relationships and associated correlation values between chl-a, TSS, total phosphate, and TKN and visible TM and OLI bands at the 26 selected stations in Lake Okeechobee. Equation (7) represents the basic form of Pearson's correlation as follows:

$$R = \frac{\sum (X_{band} - \overline{X_{band}})(Y_{WQP} - \overline{Y_{WQP}})}{\sqrt{(X_{band} - \overline{X_{band}})^2 + (Y_{WQP} - \overline{Y_{WQP}})^2}} \tag{7}$$

$X_{band}$ = the value of corrected reflectance, $\overline{X_{band}}$ = the mean value of the corrected reflectance, $Y_{WQP}$ = WQPs data from monitoring stations, and $\overline{Y_{WQP}}$ = the mean value of the in situ WQP.

Data from monitoring stations.

A linear multiple regression analysis for all four WQPs conducted and a general formula for each index obtained as follows:

$$WQP = a + (b \times X_{k,1}) + (c \times X_{k,2}) + (d \times X_{k,3}) + (e \times X_{k,4}) \tag{8}$$

In this equation, WQP is the dependent variable representing chl-a, TSS, total phosphate and TKN at each monitoring station of k, and X represents the independent reflectance variable calculated from above-mentioned equations for Landsat-5/TM or Landsat-8/OLI data at each monitoring station of k. The numbers indicated the band number for each sensor and a, b, c, d, and e are the model coefficients based on the least square algorithm using both measured WQP value at each station and the known pixel reflectance values.

## 3. Results and Discussion

### 3.1. Chlorophyll-a and TSS

The observed electromagnetic properties of chl-a and TSS for this study were in accordance with those previously established in the literature. The prominent scattering-absorption features for chl-a is strong absorption between 450 and 475 nm (blue) and at 670 nm (red), and reflectance peaks at 550 nm (green) and near 700 nm (NIR). This

reflectance peak near 700 nm and its ratio to the reflectance at 670 nm have been used in other studies to develop algorithms and was present in this analysis.

Pearson's correlation was carried out to detect the existence of significant relationships between Landsat bands with chl-a and TSS in-situ concentrations. The independent variables were all the visible bands and their ratios presented in Table 2. Variables with $p > 0.10$ were removed from the analysis.

**Table 2.** Pearson's R correlation between limnological data and Landsat bands and ratios.

| Bands/Band Ratios | Dry Season | | Wet Season | |
|---|---|---|---|---|
| | Chl-a (mg/m$^3$) | TSS (mg/L) | Chl-a (mg/m$^3$) | TSS (mg/L) |
| Blue (B) | −0.70 ** | −0.29 | −0.45 ** | −0.55 ** |
| Green (G) | −0.60 ** | −0.27 | −0.31 ** | −0.35 ** |
| Red (R) | −0.80 ** | −0.07 | −0.42 ** | −0.41 ** |
| Near Infrared (NIR) | −0.63 ** | 0.20 | −0.24 | −0.32 ** |
| B/G | 0.26 | −0.27 | −0.23 | −0.15 |
| B/R | 0.78 ** | −0.36 ** | 0.20 | 0.05 |
| B/NIR | 0.60 | −0.47 ** | −0.30 ** | −0.36 ** |
| G/B | −0.25 | 0.25 | 0.14 | 0.06 |
| G/R | 0.82 ** | −0.22 | 0.76 ** | 0.64 ** |
| G/NIR | 0.54 | −0.46 ** | −0.12 | 0.49 ** |
| R/B | −0.76 ** | 0.32 | −0.28 | −0.29 |
| R/G | −0.81 ** | 0.22 | −0.76 ** | −0.68 ** |
| R/NIR | −0.06 | −0.49 ** | −0.43 ** | −0.62 ** |
| NIR/B | −0.56 | 0.43 ** | −0.01 | −0.06 |
| NIR/G | −0.51 | 0.46 ** | −0.01 | −0.01 |
| NIR/R | 0.08 | 0.50 ** | 0.29 | 0.08 |

(**): significant correlation for $p < 0.05$.

There is a high dynamic range in the correlation analysis between Landsat bands and chl-a and TSS concentration. The two highest values of coefficient of correlation for Chl-a was found in Red/Green ratio (−0.81) and also Green/ Red ratio (0.82) in dry season, and Red/Green (−0.76) and also Green/ Red ratio (0.76) in wet season at a significance level of $p < 0.05$.

Similarly, the two notable correlation between the Landsat bands and TSS was found in Red/NIR ratio (−0.49) and also in NIR/Red ratio (0.50) in dry season, and Red/Green ratio with a coefficient of correlation (R) value of −0.68 and also Green/Red band ratio with a coefficient of correlation value of 0.64 in wet season. In wet season, the strongest correlation between TSS values and single Blue band was −0.55 and at a significance level of $p < 0.05$ was obtained.

Multiple regression models were constructed for chl-a and TSS through band and band ratios that indicated a good correlation (Table 2). The bands and band ratios of B, B/R, G/R, R/B, and R/G were selected for the estimation of chl-a in dry season, and band R, G/R, R/G, and R/NIR ratios selected for wet season. Equations (9) and (10) represent the resulted models for chl-a estimation.

$$\text{Chl} - \text{a in dry season} = a + (b \times B) + \left(c \times \frac{\text{Blue}}{\text{Red}}\right) + \left(d \times \frac{\text{Green}}{\text{Red}}\right) + \left(e \times \frac{\text{Red}}{\text{Blue}}\right) + \left(f \times \frac{\text{Red}}{\text{Green}}\right) \quad (9)$$

$$\text{Chl} - \text{a in wet season} = a + (b \times R) + \left(c \times \frac{\text{Green}}{\text{Red}}\right) + \left(d \times \frac{\text{Red}}{\text{Green}}\right) + \left(e \times \frac{\text{Red}}{\text{NIR}}\right) \quad (10)$$

For the variability of TSS in the dry season, the chosen bands and band ratios were B/NIR, G/R, R/B, and R/G. For the wet season they were B, R, G/R, R/G, and R/NIR. The chosen bands showed significant relationships with observed values in the dry ($R^2 = 0.67$) and wet season ($R^2 = 0.60$). Another criteria in selecting variables for the MLR model development was the Durbin–Watson which needs to fall between the two critical values

of 1.5 < d < 2.5, and therefore, there is no first order linear auto-correlation in our multiple linear regression data. The Durbin–Watson values of 2.249 and 1.650 were obtained in dry and wet seasons, respectively, which fall between the two critical values of 1.5 < d < 2.5, and therefore, there is no first order linear auto-correlation in our multiple linear regression data. The functional models for TSS are

$$\text{TSS in dry season} = a + \left(b \times \tfrac{\text{Blue}}{\text{NIR}}\right) + \left(c \times \tfrac{\text{Green}}{\text{NIR}}\right) + \left(d \times \tfrac{\text{Red}}{\text{NIR}}\right) + \left(e \times \tfrac{\text{NIR}}{\text{Blue}}\right) + \left(f \times \tfrac{\text{NIR}}{\text{Green}}\right) + \left(g \times \tfrac{\text{NIR}}{\text{Red}}\right) \quad (11)$$

$$\text{TSS in wet season} = a + (b \times B) + (c \times R) + \left(d \times \frac{\text{Green}}{\text{Red}}\right) + \left(e \times \frac{\text{Red}}{\text{Green}}\right) + \left(f \times \frac{\text{Red}}{\text{NIR}}\right) \quad (12)$$

Based on the best and significant relationships found for between the measured concentration of chl-a and TSS and the Landsat TM and OLI data, four regression models developed for these two parameters for each dry and wet seasons. The regression analyses for chl-a and TSS are shown in Table 3. Table 4 presents these multiple regression models constructed through Equations (9)–(12).

**Table 3.** Regression analyses for chl-a and total suspended solid (TSS).

| Season | Parameter | $R^2$ | Standard Error | *p* Value | Durbin-Watson | Observations |
|--------|-----------|-------|----------------|-----------|---------------|--------------|
| Dry | Chl-a | 0.84 | 8.84 | 0 | 1.797 | 48 |
|  | TSS | 0.67 | 6.07 | 0 | 2.249 | 48 |
| Wet | Chl-a | 0.48 | 20.88 | 0.005 | 2.103 | 38 |
|  | TSS | 0.6 | 12.77 | 0.002 | 1.65 | 38 |

**Table 4.** Regression analysis between Landsat reflectance and limnological parameters.

| Season | Water Quality Parameters | Regression Equations Derived |
|--------|--------------------------|------------------------------|
| Dry | Chl-a (mg/m³) | $= 881.1 \times (B/R) - 1784.7 \times (G/R) + 5331.5 \times (R/B) - 3096.2 \times (R/G) \times 1167 \times (B) + 525.5$ |
|  | TSS (mg/L) | $= 517.91 \times (R/NIR) - 8.86 \times (G/NIR) - 799.23 \times (NIR/B) + 127.76 \times (NIR/G) + 1100.92 \times (NIR/R) - 74.63 \times (B/NIR) - 1037.79$ |
| Wet | Chl-a (mg/m³) | $= -1067.77 \times (G/R) - 2144.45 \times (R/G) - 35.04 \times (R/NIR) + 297.75 \times (R) + 3095.6$ |
|  | TSS (mg/L) | $= 361.89 \times (R) - 1018.25 \times (G/R) - 1919.21 \times (R/G) - 26.15 \times (R/NIR) - 182.90 \times (B) + 2855.76$ |

The R square values proved the accuracy of reliability of first four bands of Landsat images and their ratios for the measurement of chl-a and TSS concentrations at studied waterbody.

To visualize the models created in Table 4, maps shown in Figure 2 were created for the entire area of lake Okeechobee. Each map was made from the indicated date and a single image.

Based on the standardized residuals normal probability–probability (P-P) plots were created. Part of the data from monitoring stations were used for validation of the satellite based water quality estimates from the different images which were also used to develop the P-P plots and also to check the normality assumptions (Figure 3). Figure 3 shows that Chl-a in dry season show a good performance indicating the cumulative distribution function (CDF) of both the observed and modeled values closely matches. For the wet season, the CDF of the modeled value distribution is higher than that of the observed one for some of the values.

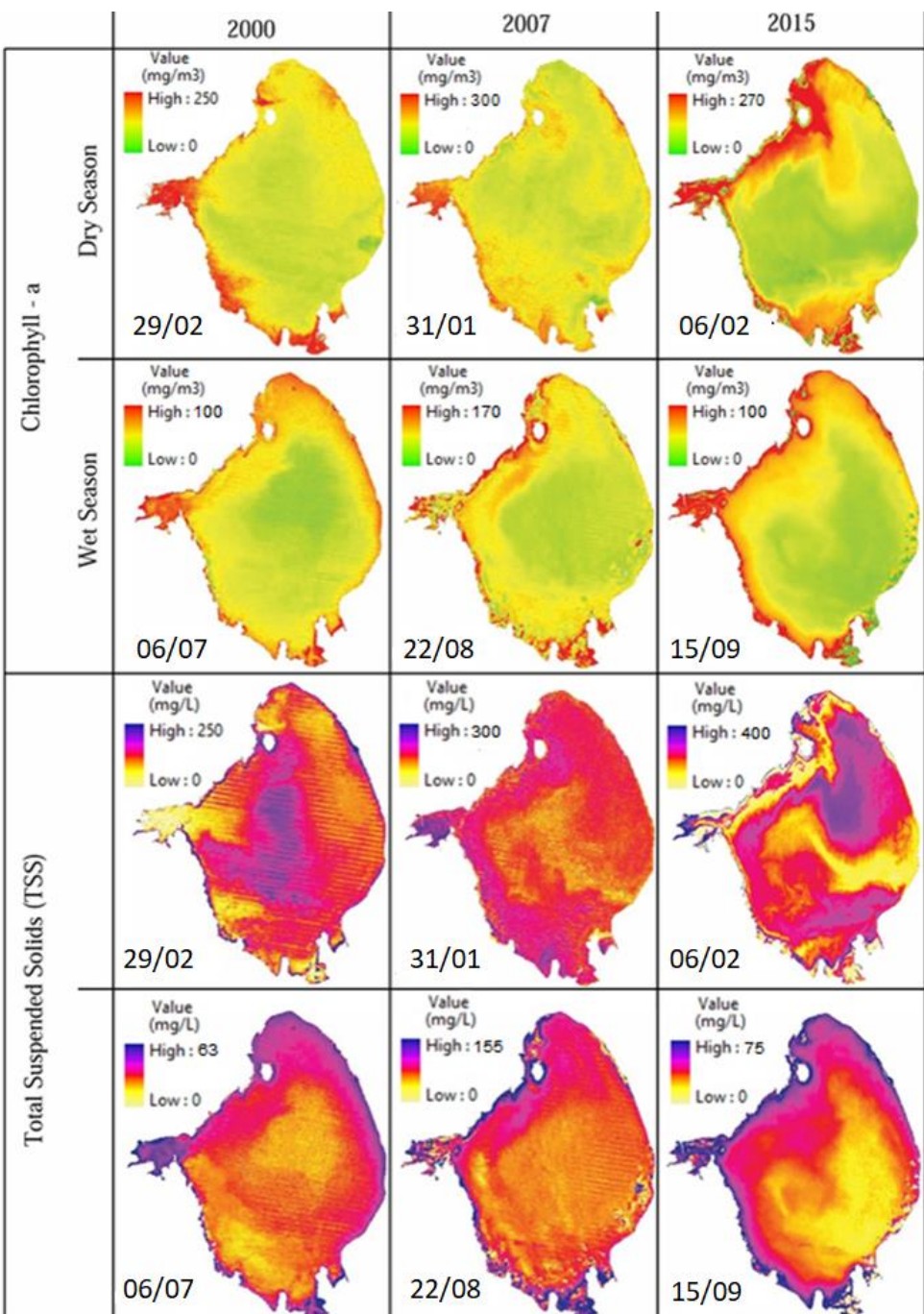

**Figure 2.** Predicted seasonal and spatial chl-a and TSS values in Lake Okeechobee.

*3.2. Nutrients*

Based on the relationship between nutrients, chl-a or TSS, and also water quality monitoring stations' data and Landsat bands and ratios, Pearson correlations calculated and are summarized in Table 5. Based on these results, four models were developed for total phosphate and TKN in two different dry and wet seasons.

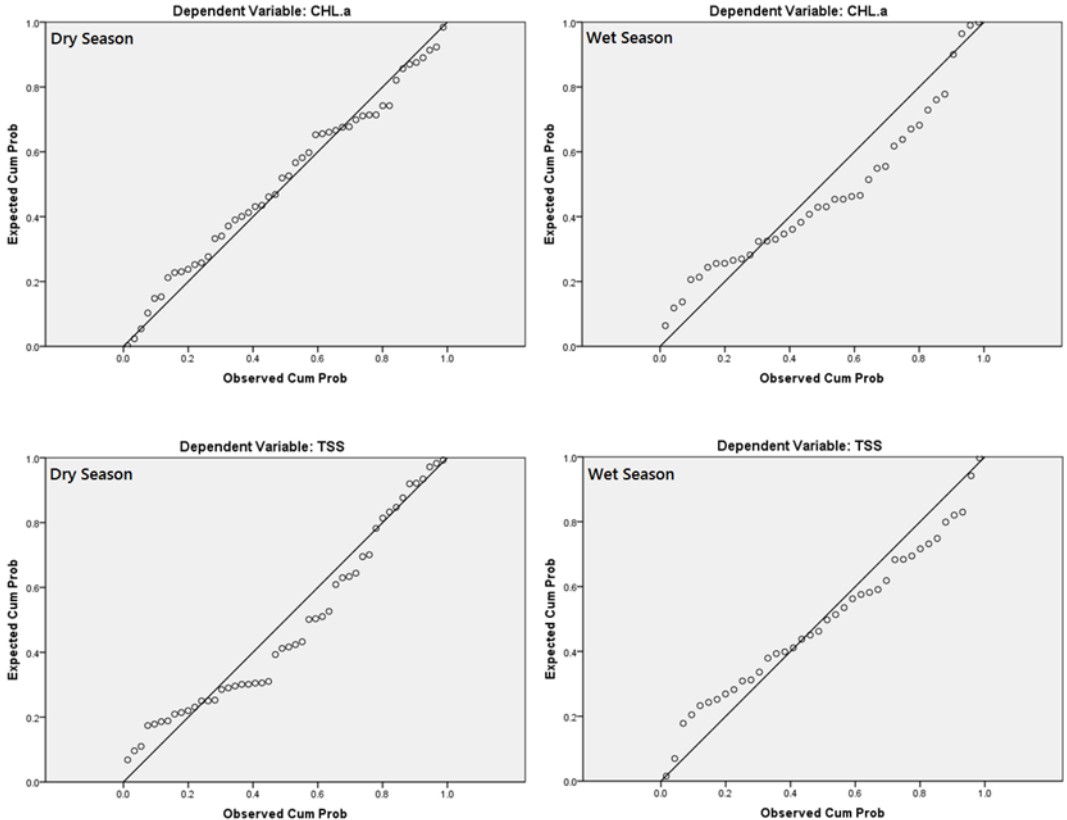

**Figure 3.** The normal probability-probability (P-P) plots of regression standardized residuals for total phosphate and Chl-a and TSS in wet and dry seasons.

**Table 5.** Pearson's R correlation for the in-situ water quality parameters and Landsat bands and ratios.

| Bands | Season | | Season | |
|---|---|---|---|---|
| | **Dry** | **Wet** | **Dry** | **Wet** |
| | **TP (mg/L)** | | **TKN (mg/L)** | |
| B | 0.34 ** | 0.59 ** | −0.36 ** | 0.01 |
| G | 0.35 ** | 0.55 ** | −0.12 | 0.03 |
| R | 0.50 ** | 0.59 ** | −0.31 ** | 0 |
| NIR | 0.31 ** | 0.61 ** | 0.03 | 0.08 |
| B/G | −0.21 | −0.17 | 0.19 | −0.05 |
| B/R | −0.52 ** | −0.40 ** | 0.19 | 0.05 |
| B/NIR | −0.28 | −0.54 ** | 0.07 | −0.29 ** |
| G/B | 0.21 | 0.2 | −0.19 | 0.07 |
| G/R | −0.53 ** | −0.58 ** | 0.03 | 0.29 ** |
| G/NIR | −0.21 | −0.62 ** | −0.1 | −0.32 ** |
| R/B | 0.52 ** | 0.47 ** | −0.17 | −0.08 |
| R/G | 0.52 ** | 0.60 ** | −0.04 | −0.29 ** |
| R/NIR | 0.19 | −0.52 ** | −0.15 | −0.49 ** |
| NIR/B | 0.27 | 0.62 ** | −0.06 | 0.21 |
| NIR/G | 0.21 | 0.66 ** | 0.07 | 0.25 |
| NIR/R | −0.2 | 0.59 ** | 0.14 | 0.42 ** |
| Chl-a (mg/m$^3$) | −0.02 | 0.47 ** | 0.51 ** | 0.91 ** |
| TSS (mg/L) | 0.90 ** | 0.57 ** | 0.79 ** | 0.73 ** |

(**): significant correlation for $p < 0.05$.

Total phosphate was found to be highly correlated with TSS, and acceptably tied to NIR band, and Green/Red band ratio in dry season, and correlated with TSS, Blue/Red,

NIR/Green, and NIR/Red in wet season. The developed models for total phosphate concentration in dry and wet seasons are as follow:

$$\text{TP in dry season} \ = \ a \ + \ (b \ \times \ TSS) \ + \ (c \ \times \ \frac{\text{Green}}{\text{Red}}) \ + \ (d \ \times \ NIR) \qquad (13)$$

$$\text{TP in wet season} \ = \ a \ + \ (b \ \times \ TSS) \ + \ (c \ \times \ \tfrac{\text{Blue}}{\text{Red}}) \ + \ (d \ \times \ \tfrac{\text{NIR}}{\text{Green}}) \ + \ (e \ \times \ \tfrac{\text{NIR}}{\text{Red}}) \qquad (14)$$

The variability of TKN concentration was also investigated using the same procedure, incorporating visible bands and their ratios as independent variables in the regression analysis. From the stepwise variable selection procedure, the following functional model was selected in dry and wet season:

$$\text{TKN in dry season} = a + (b \times Chl\text{-}a) + (c \times TSS) + (d \times B) + (e \times R) \qquad (15)$$

$$\text{TKN in wet season} \ = \ a \ + \ (b \ \times \ Chl-a) \ + \ (c \ \times \ TSS) \ + \ (d \ \times \ \tfrac{\text{Blue}}{\text{NIR}}) \ + \ (e \ \times \ \tfrac{\text{Green}}{\text{NIR}}) \ + \ (f \ \times \ \tfrac{\text{Red}}{\text{NIR}}) \ + \ (g \ \times \ \tfrac{\text{NIR}}{\text{Red}}) \quad (16)$$

The statistical values given in Table 6 were obtained from the regression analysis computation. Based on the calculated R square values, it was found that the first four bands of Landsat images and their ratios and the concentrations of chl-a and TSS in the Lake Okeechobee can be reliably used for the measurement of total phosphate and TKN in this lake.

**Table 6.** Regression analyses for TP and TKN.

| Season | Parameter | $R^2$ | Standard Error | Durbin-Watson | Observations |
|---|---|---|---|---|---|
| Dry | TP | 0.92 | 0.015 | 1.974 | 50 |
| | TKN | 0.94 | 0.097 | 2.027 | 50 |
| Wet | TP | 0.89 | 0.025 | 2.488 | 38 |
| | TKN | 0.93 | 0.166 | 1.627 | 48 |

It was found that TP was correlated with TSS (R = 0.90) and also with G/R band ratio (R = −0.53) in dry season, and in wet season found to be highly correlated with G/NIR band ratio (R = −0.62) and NIR/G band ratio (R = 0.66) (at a significance level of $p < 0.05$ (Table 5). Multiple linear regression models developed for total phosphate estimation showed high coefficient of determination equal to 0.92 and 0.89 in dry and wet seasons, respectively (Table 6). The developed MLR models for the estimation TP and TKN using both the Landsat data and in situ measurement data. The Durbin–Watson values of 1.947 and 2.488 in dry and wet seasons, respectively, determines that there was no first order linear auto-correlation in the resulted multiple linear regression models.

The top two significant correlation found between the Blue band (R = −0.36) and TSS (R = 0.79) and TKN were in the dry season. In wet season, TKN was correlated with R/NIR band ratio (R = −0.49) and chl-a (R = 0.91) at a significance level of $p < 0.05$ (Table 5). Chl-a and TSS, and highest correlated bands and band ratios were selected to develop multiple regression models for TKN, which are summarized in Table 7. The obtained values Durbin–Watson equal to 2.027 and 1.627 for dry and wet seasons, respectively, shows that there is no first order linear auto-correlation in the developed multiple linear regression models. Figure 4 shows different spatial and temporal distribution of total phosphate and TKN over the entire body of Lake Okeechobee, resulted from the developed equations represented in Table 7.

**Table 7.** Regression equations for TP and TKN, as a function of Landsat bands and other water quality parameters.

| Season | Water Quality (mg/L) | Regression Equations Derived |
|---|---|---|
| Dry | TP | $= 0.001(TSS) - 0.202(G/R) - 2.56(NIR) + 0.468$ |
| | TKN | $= 0.008(TSS) + 0.009 (Chl\text{-}a) + 3.91(B) - 4.35(R) + 0.641$ |
| Wet | TP | $= 0.002(TSS) + 0.154(B/R) + 1.66(NIR/G) - 1.23(NIR/R) - 0.232$ |
| | TKN | $= 0.017(Chl\text{-}a) + 0.001(TSS) - 0.057(B/NIR) + 0.345(G/NIR) - 1.09(R/NIR) - 0.249(NIR/R) + 2.21$ |

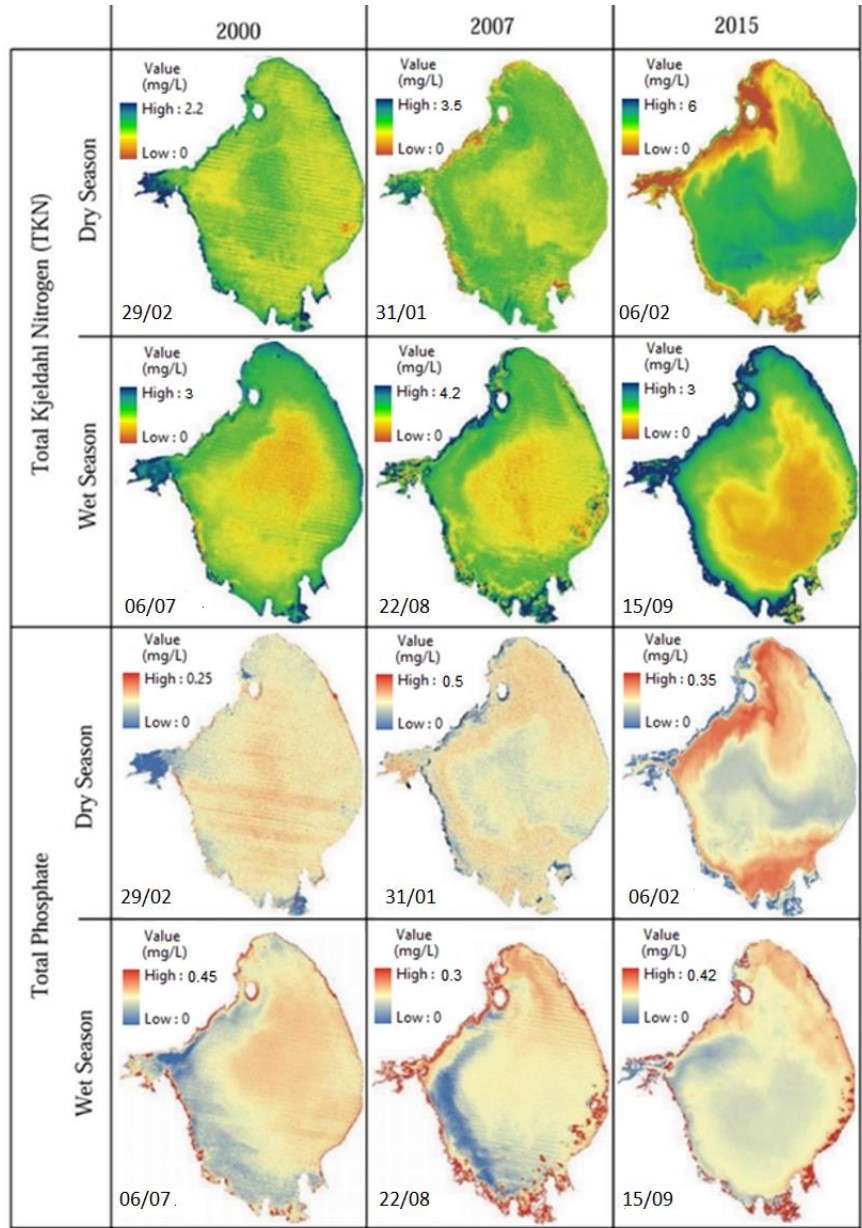

**Figure 4.** Seasonal and spatial dynamics of TP and TKN in Lake Okeechobee.

The normal probability–probability (P-P) plots for the studied variables in both dry and wet seasons indicated the normal distribution for the total phosphate and TKN in both dry and wet seasons (Figure 5). The 25% of data from monitoring stations were used for validation and tested these against the extracted values from arbitrarily selected points in the area of stations and different used images to develop the P-P plots and also to check the normality assumptions. Figure 5 shows that, except for TKN in the wet season, the

CDF values for both observed and modeled TKN and TP are similar indicating a good correspondence in the distribution of the modeled and monitored values.

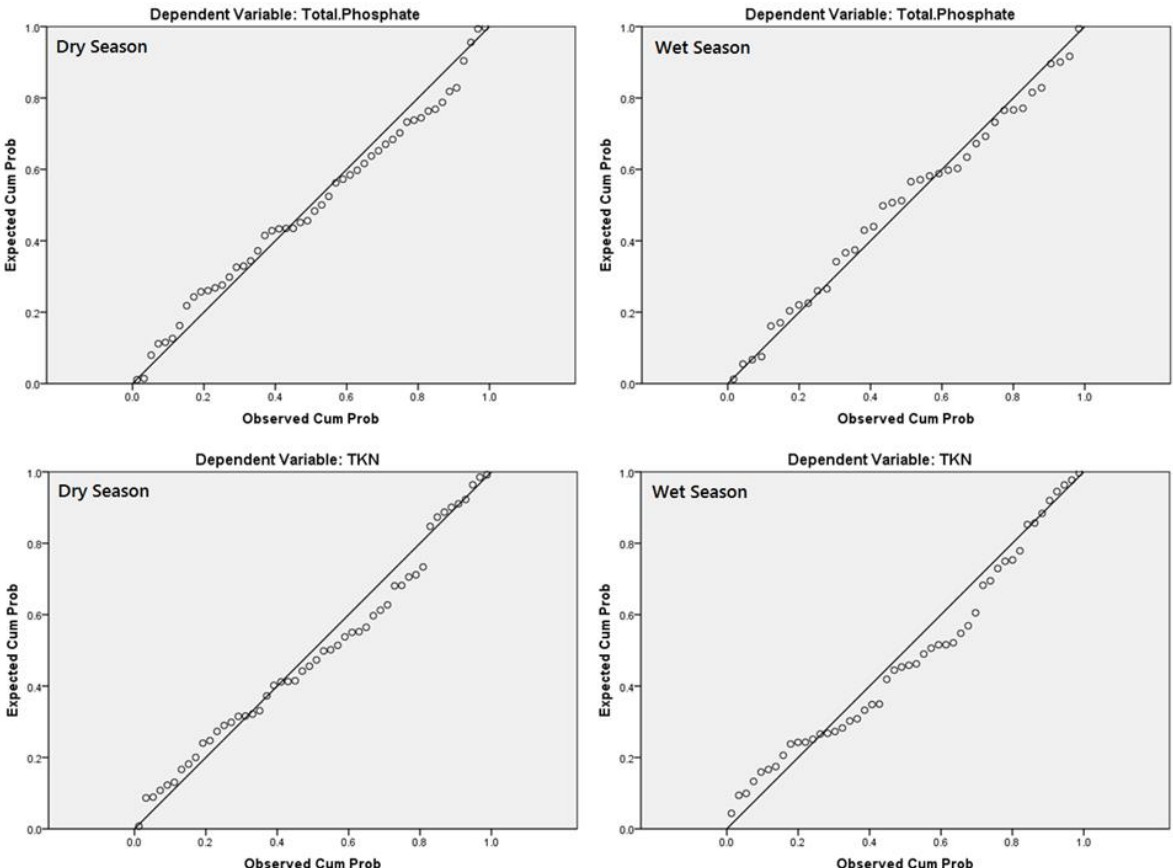

**Figure 5.** The normal probability–probability (P-P) plots of regression standardized residuals for TP and TKN in dry and wet seasons.

Using the spatial maps of water quality parameters shown in Figures 2 and 4, different classes showing ranges of values for each parameter were developed and the corresponding area of the lake falling in each class is shown in km². This is shown in Table 8 for eth two seasons and the three years (2000, 2007, and 2015).

**Table 8.** The area-based condition of water quality parameters in Lake Okeechobee in dry and wet seasons, and in three years of 2000, 2007, and 2015 (km²) in the ranges of selected values.

| | | **Dry Season** | | | | | | **Wet Season** | | | | |
|---|---|---|---|---|---|---|---|---|---|---|---|---|
| | Year | 0–50 | 50–100 | 100–150 | 150–200 | >200 | | 0–20 | 20–40 | 40–60 | 60–80 | 80–100 | >100 |
| Chl-a | 2000 | 51.6 | 287.3 | 722.9 | 175.2 | 93.1 | | 514.7 | 842.9 | 50.5 | 7 | 0.1 | 0 |
| (mg/m³) | 2007 | 96.2 | 334.1 | 451 | 317.9 | 170.2 | | 616.7 | 687.8 | 63.8 | 37.9 | 3.6 | 5.4 |
| | 2015 | 117.4 | 226.6 | 767.3 | 247.9 | 8 | | 913.5 | 346.2 | 80.6 | 66.1 | 8.8 | 0 |
| | Year | 0–40 | 40–80 | 80–120 | 120–160 | >160 | | 0–15 | 15–30 | 30–45 | 45–60 | 60–75 | >75 |
| TSS | 2000 | 31.9 | 17.9 | 543.9 | 735.8 | 85.4 | | 605.4 | 772.2 | 31.5 | 6 | 0.1 | 0 |
| (mg/L) | 2007 | 46.9 | 74.2 | 389.9 | 845.2 | 58.5 | | 555.8 | 739.8 | 67.1 | 42.4 | 3.5 | 6.3 |
| | 2015 | 149.8 | 154.2 | 222.2 | 253.3 | 637 | | 932.7 | 359.2 | 75.8 | 45.4 | 2.4 | 0 |
| Total Phos-phate (mg/L) | Year | 0–0.07 | 0.07–0.15 | 0.15–0.20 | 0.20–0.25 | 0.25–0.30 | >0.30 | 0–0.07 | 0.07–0.15 | 0.15–0.20 | 0.20–0.25 | 0.25–0.30 | >0.30 |
| | 2000 | 5.9 | 431.3 | 968 | 6.8 | 0.6 | 2 | 211.5 | 1166 | 22.3 | 5.9 | 3.5 | 6 |
| | 2007 | 57.4 | 530.5 | 627.4 | 178.1 | 18.5 | 3 | 209.8 | 997 | 45.9 | 23.3 | 20.7 | 117.8 |
| | 2015 | 71.7 | 15.8 | 885.3 | 442.5 | 0.01 | 0 | 104.5 | 1244 | 9.6 | 6.1 | 5.9 | 45.2 |
| | Year | 0–1.0 | 1–1.5 | 1.5–2.0 | 2.0–2.7 | 2.7–3.5 | >3.5 | 0–1.0 | 1–1.5 | 1.5–2.0 | 2.0–2.7 | 2.7–3.5 | >3.5 |
| TKN | 2000 | 6.7 | 1336 | 58.4 | 10.2 | 2.2 | 2 | 280.1 | 794 | 313 | 23.8 | 3.6 | 0 |
| (mg/L) | 2007 | 264.7 | 963.4 | 138.1 | 11.2 | 0.3 | 0 | 300.8 | 683 | 339 | 86.6 | 1.2 | 4.7 |
| | 2015 | 0.1 | 0.2 | 110.7 | 697.7 | 510.6 | 5 | 709.4 | 432 | 152 | 103 | 18.6 | 0 |

It is shown that large area of the lake has Landsat based Chla-a values in the range of 150–200 mg/m$^3$ in the dry season and in the range of 0–40 mg/m$^3$ in wet season (Table 8).

TSS values are in the range of 120–160 and 0–15 mg/L in the dry and wet seasons, respectively. Large areas of the lake in the three years showed TP values in the ranges of 0.15–0.2 and 0.07–0.15 mg/L for the dry and wet seasons, respectively. Similarly, TKN values fall in the range of 1–1.5 mg/L in 2000 and 2007 for the dry as well as wet seasons.

## 4. Conclusions

Watershed processes have a direct impact on the water quality of receiving water bodies including lakes. Understanding the dynamics of water quality in lakes will require a monitoring program that is feasible, reliable, and that can provide accurate results with acceptable spatial and temporal scales. This will be a challenge, especially in large lakes unless field observations are coupled with remote sensing approaches.

The spatiotemporal variability of water quality parameters along with the seasonal changes were studied for Lake Okeechobee, South Florida using remote sensing and in-situ water quality data. The dynamics of selected four parameters: total phosphate (TP), total Kjeldahl nitrogen (TKN), total suspended solid (TSS) and chlorophyll-a (chl-a) were analyzed using data from satellites and water quality monitoring stations. Statistical approaches were used to establish correlation between reflectance data and observed water quality records. Stepwise MLR approach was used to establish correlations between reflectance data and the selected water quality parameters. The predictive stepwise MLR models to estimate chl-a and TSS depicted R square values of 0.84 for chl-a and 0.67 for TSS in dry season and moderate R square values of 0.48 for chl-a and 0.60 for TSS in wet season.

The MLR analysis indicated that for the dry season, TP and TKN were correlated with Landsat reflectance data with R square values of 0.92 and 0.94, respectively. The wet season analysis also showed a strong correlation for TP and TKN with 0.89 ad 0.93 R square values, respectively.

Results of this study showed that the application of remote sensors like TM and LOI from Landsat are useful to monitor large lakes by providing reflectance data that can be used to develop predictive models for estimating the dynamics of water quality parameters.

**Author Contributions:** Conceptualization, M.H., A.M.M.; methodology, M.H.; software, M.H.; validation, M.H.; formal analysis, M.H.; investigation, M.H.; resources, M.H., A.M.M.; data curation, M.H.; writing—original draft preparation, M.H.; writing—review and editing, M.H., A.M.M., A.M., S.K.; visualization, M.H., A.M.M.; supervision, A.M.M.; project administration, A.M.M.; funding acquisition, M.H. All authors have read and agreed to the published version of the manuscript.

**Funding:** This research received no external funding.

**Data Availability Statement:** Not applicable.

**Acknowledgments:** The research was funded by the Florida International University, Miami, USA. The observational data were obtained from South Florida Water Management District (SFWMD). We thank the reviewers and the journal officials for providing insightful comments.

**Conflicts of Interest:** The authors declare no conflict of interest.

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
