# Peer review of "Land–Lake Linkage and Remote Sensing Application in Water Quality Monitoring in Lake Okeechobee, Florida, USA"

_land, doi:10.3390/land10020147_

Round 1

Reviewer 1 Report

This manuscript develops remote sensing methods for water quality parameters in Lake Okeechobee. This study builds on a lot of others from different locations around the globe, but the relatively novel part of this manuscript is the development of methods for total phosphate and nitrate. The authors present a thorough analysis of field observations with satellite reflectance data to develop remote sensing methods. The general approach is sound, however a lot of the details are missing and there’s no validation of the developed methods, so it’s hard to fully evaluate the methods (see details below). That said, I think the results of this study could be really useful and is of interest to the wider community as it will enable detection of water quality parameters from Landsat.

General Comments:

1) In the methods you state the field data was split into a development and validation dataset. Where are the results of the validation? It seems like only the development of the method/s were presented in this manuscript. For a robust remote sensing method, you need to show a validation on a different set of data.

2) You use the field chl-a and TSS observations in the MLR to develop relationships to predict TP and TKN, and end up with really good R2 values on these relationships. But in practice, you’re going to use satellite derived chl-a and TSS, which adds a layer of uncertainty. How well do your relationships work in that case? This is connected to my previous point: where’s the validation? If you have some of the field data for validation, then you can check how well your nutrient algorithms work when you use satellite reflectance data to first derive chl-a and TSS and then TP and TKN. Again, this would demonstrate the robustness of your methods.

3) There are a few places in the manuscript the authors state they present “spatiotemporal variability of water quality parameters”. I would disagree. The authors develop methods they apply to a handful of Landsat images (3 images per season, separated by 7 years). These give an idea of the kind of spatiotemporal variability that might exist, but there is no discussion or interpretation of these maps and data. If the authors want to do analysis of the spatiotemporal variability, then they need to include a lot more satellite data, as the daily images they currently present might be anomalous. It’s hard to tell without more data. However, I don’t necessarily think the authors need to do that for this manuscript. I think showing their methods applied to a handful of images is fine, but then they shouldn’t say they are doing spatiotemporal variability analysis. In addition, I think some discussion should be included about the maps they produce. Are the lake-wide patterns reasonable/expected given the time of year?

4) There are a lot of formatting inconsistences/problems in tables (some parts in bold, some not) and equations/symbols.

Specific Comments:

L184-188 These sentences feel out of place here and more suited to the Introduction. The authors have already described these connections, so it doesn’t need to be repeated again here.

Section 2.2.1 – this section is missing some details which makes it hard to assess the methods used here. Plus, as the method description currently stands, this study would be almost impossible to reproduce/replicate.  Specifically:

1) Can you please provide a link to the data repository where you accessed the data?

2) There is no information on the collection of the samples, how they were processed, what depths they come from, etc. Can you please include some of these details in the manuscript (and even refer to any documentation of the SFWMD that describes these)?

3) Can you include details on the method you used to split your dataset for development and validation? Was it done randomly, based on some criteria?

4) The last paragraph in this section states a list of different “data pre-treatment methods”. Can you please include more details of these in the manuscript?

Section 2.2.2 – similarly to section 2.2.1, I find some of the details are missing here.

1) Why did you only select data from those three years and on those specific dates? Were there not more dates with clear scenes of Lake Okeechobee?

2) Why did you only use those two Landsat satellites? Why none of the others?

3) In this section you mention “separate methodologies” to do the atmospheric correction and radiometric calibration of TM and OLI data, but don’t say anything more about them. I originally thought you hadn’t included the details but you do in the next section. I’d recommended stating in here somewhere “details in Section XX”.

Section 3. Methodology – I think this section should be combined with Section 2. Materials and Methods.

Section 3.1. This section is challenging to follow. Partly, it’s because I think there is a problem with the formatting of the equations and the symbols in the text in the version of the PDF I’m reviewing. But also, I think some more details are needed in the text. It reads as one equation after another, with each symbol in each equation described, but not real connection from one equation to the next. As I read this section, the questions that come to mind are: What parameter/level of data did you acquire from USGS? Did you get the raw digital numbers and do all this processing yourself? If so, why? Are there new steps in here? Can you not download surface reflectances directly? If not, then I don’t think you need to include all these details. You can just state the version of the data you used and refer the reader to the specific atmospheric correction texts. Also, what is the difference between the TM and the OLI processing? That isn’t clear to me from this section.

Eq7 – Again, I think there’s a problem with the formatting here and the accompanying text. I’m reading this as Xband – Xband and Y_WQP – Y_WQP (which doesn’t make any sense).

L263-265: These sentences are out of place here. They’ve already been stated earlier in the Introduction.

L294: suggest removing the “which were included in the multiple linear regression” from this sentence. It’s a bit misleading as it makes the reader think Table 2 has something to do with the multiple linear regression results, which it doesn’t.

L306-308: suggest moving this paragraph to after the details of the multiple regression models in the next 2 paragraphs. That way you’re describing the models, then presenting the results (like you do in section 4.2 with the nutrient analysis). On first reading, I was looking at Table 3 wondering what bands and band ratios you’d used to do the regressions.

L313: Can you please include the details on what criteria you used to indicate a “good correlation” and hence select the bands for the multiple regressions? For example, for chl-a in the dry season, one of the highest correlation coefficients is with the red band but you don’t use it in the regression model? Similarly for section 4.2 and the nutrient correlations, you select the NIR for regressing against total phosphate, even though there is a higher correlation with the red band.

Do you not expect the inverse band ratios to have approximately inverse correlations? e.g. if R = 0.6 for B/G, it should be approximately -0.6 for G/B because B/G is the inverse of G/B. Therefore, when you’re doing your multiple linear regression are you not effectively “weighting” each fit by the ratios?

Figure 2 & 4: Can you please include more details on these maps? Are they made from one image within each season? Averages from all the images? The color bars are hard to read, can you please make the text larger?

Figure 3 & 5: Caption says it shows total phosphate as well as chl-a and TSS. Again, can you please include more details on which images/maps these plots came from? One of the ones shown in Fig 2? Some average map?

L357-375: There’s a lot of mention of R^2 values in this section. Are these not R values (rather than R^2 values)?

L366: Do you not mean R^2 (i.e. coefficient of determination) rather than “correlation coefficients”

Table 8: Part of this table is cut off/runs off the edge of the page. Also, what are the different columns e.g. 0-50, 50-100, etc?

Typos, etc:

L101 “found” not “founds”

L102 “plant” not “plants”

L120-103 “which their increase” should be “and their increase”

L103-106 this sentence is a bit long-winded, suggest splitting it up into two sentences.

L106 “finding” not “founding”

L110 there’s an extra “.” that isn’t needed (“and.”)

L111 I assume TM1 = TM Band 1, etc. Can you please explain this notation explicitly in the text the first time you use it?

L145 “introduced” not “introduces”

L143-146 was this application to Landsat 7 data in Mersin Bay part of the Gong et al study? If so, can you please make this clearer in the text? If not, can you provide the citation?

Fig 1: the bottom of the words “US” and “Florida” are cut off – can you please correct this?

L278/279: this should all be in one paragraph.

L334 the “the” in “residuals the normal” isn’t needed

L335 there’s an extra “.” after “created”

L335 “which” isn’t needed

L335: I think you just mean figure 3? (Not figure 4 too)

L368-370: this sentence doesn’t make sense. I think you can remove it.

Ref 30 missing

Reviewer 2 Report

English grammar needs to be checked and a proper register needs to be used in order to be in accordance with the journal's standards. This coud be accomplished by a native English speaker.

Reviewer 3 Report

It is a very well structured and well-written manuscript emphasizing on remote sensing application for water quality. There are few minor issues:

Line 17: …  dry and west seasons were used … (please correct misspelled world)

Line 47: Please add following paragraphs right before line 47: Advanced geostatistical techniques are gaining support among researchers for spatiotemporal evaluation of surface water quality (Jat et al. 2018) when space/time monitoring data are sparse. However, data collection is an expensive and time-consuming process. Spatial and temporal ….. (line 47).

Line 116: …the concentration t of total.. (remove ‘t’)

Figure 1: Legend box is smaller than legends (please enlarge legend box to fit legends)

Table 8: Portion of the table is outside the paper margin

Results and discussion Part:

Please add brief explanation on:  why predicted chlorophyll–a in both wet and dry seasons is higher (red) around lake border areas (littoral zone) whereas TSS is lower in same area (specifically in dry season of 2015)? Is it boundary effect (boundary effect on reflectance)?  

Reference:

Jat, P., Serre, M.L. A novel geostatistical approach combining Euclidean and gradual-flow covariance models to estimate fecal coliform along the Haw and Deep rivers in North Carolina. Stoch Environ Res Risk Assess 32, 2537–2549 (2018). https://doi.org/10.1007/s00477-018-1512-6

Love, D.C., Lovelace, G.L., Money, E.S. /et al./ Microbial Fecal Indicator Concentrations in Water and Their Correlation to Environmental Parameters in Nine Geographically Diverse Estuaries.
/Water Qual Expo Health/ *2, *85–95 (2010). https://doi.org/10.1007/s12403-010-0026-3

Round 2

Reviewer 1 Report

Thanks to the authors for responding to my comments and suggestions. I have a few outstanding and further comments:

Section 2.2.1 – I acknowledge the authors response that there was no information about collection methods, and ancillary data (e.g. depth). However, the additional details the authors provide are not really informative on the actual collection and measurement protocols and I suggest removing.

After every equation, can you please add the word “where” before you list what each symbol represents e.g. where, L lambda = …

L199: Include somewhere here that the dataset was split randomly.

L210-216: You still don’t explicitly state anywhere what parameter or level of data you are downloading. Is it the raw digital numbers? Is it surface reflectance? This should be included in this section.

L220: radiance or reflectance? I think this should be reflectance.

L253: there’s two full stops at the end of the sentence. Please remove one.

L253: Suggest rewording to “Equation 2 shows the final reflectance determined using a simplified model of atmosphere effects”

L254: remove “the”

L257: Pp should be rho p

L280: suggest adding at the end of that paragraph, “The path radiance (i.e. the upwelling atmospheric spectral radiance scattered in the direction of the sensor entrance pupil and within the sensor’s field of view), L lambda_haze, is calculated as follows:”

L287: suggest adding at the end of the sentence “where:”

Equation numbers are mixed up. The start at (1) again in section 2.3.3, rather than (7)

The equation for the pearson correlation coefficient is still incorrect. The bar is missing over the top of one of the Y_WQPs.

In the PDF that I have, in the description of the symbols in lines 305-307, the bars over the top of the symbols are misaligned i.e. it appears over the word before the symbol, rather than the symbol itself.

L348-350: In my previous review I suggested moving this paragraph to after the details of the multiple regression models in the next 2 paragraphs. In their response the authors said they had, but it hasn’t been moved. I stand by my previous comment that this isn’t the logical place for this paragraph because you haven’t presented the models yet.

L352 – 363: Somewhere in this section can you please include the explanation you gave me in your response about using the Durbin-Watson value criteria as well as the correlation coefficient values in ultimately deciding your model? Thank you for that explanation – I understand what you are doing now, but it wasn’t clear in the manuscript and I think it would be good to include it.

L372: You only include year in the figure, not exact date.

L374: Sentence “About 75%...actual data” is a bit misleading here because it makes the reader think you are working with 75% of the data, not 25%. I suggest removing it.

L376 – 379: This is a long sentence, suggest shortening it to “The validation data (25% of the full dataset) were used to evaluate the models, with P-P plots shown in Figure 3.” (If you mention the random splitting of the dataset in the methods, you don’t need to again here.)

Can you add a couple sentence discussion on the performance of the models based on the P-P plot results? Can you also please include some kind of quantitative statistics on the performance of the models e.g. R^2, mean absolute error, RMSE?

The above two comments also are relevant to the results in the TP and TKN section.

Table 8: I still don’t understand what the different columns e.g. 0-50, 50-100, etc represent.

Author Response

see attached

This manuscript is a resubmission of an earlier submission. The following is a list of the peer review reports and author responses from that submission.

Round 1

Reviewer 1 Report

The paper is clear and innovative. Table 8 does not fit perfectly in the page, and reference sources were not printed, so I suggest to check tem as I could not do it.